# 25-Hydroxycholesterol-Induced Oxiapoptophagy in L929 Mouse Fibroblast Cell Line

**DOI:** 10.3390/molecules27010199

**Published:** 2021-12-29

**Authors:** Jae-Seek You, HyangI Lim, Jeong-Yeon Seo, Kyeong-Rok Kang, Do Kyung Kim, Ji-Su Oh, Yo-Seob Seo, Gyeong-Je Lee, Jin-Soo Kim, Heung-Joong Kim, Sun-Kyoung Yu, Jae-Sung Kim

**Affiliations:** 1Departments of Oral and Maxillofacial Surgery, School of Dentistry, Chosun University, Gwangju 61452, Korea; applit375@chosun.ac.kr (J.-S.Y.); jsoh@chosun.ac.kr (J.-S.O.); 2Institute of Dental Science, School of Dentistry, Chosun University, Gwangju 61452, Korea; qjqtjdgod@naver.com (H.L.); sj23850126@hanmail.net (J.-Y.S.); kkr@chosun.ac.kr (K.-R.K.); kdk@chosun.ac.kr (D.K.K.); hjbkim@chosun.ac.kr (H.-J.K.); sky@chosun.ac.kr (S.-K.Y.); 3Department of Oral and Maxillofacial Radiology, School of Dentistry, Chosun University, Gwangju 61452, Korea; moresys@chosun.ac.kr (Y.-S.S.); hidds@chosun.ac.kr (J.-S.K.); 4Department of Prosthodontics, School of Dentistry, Chosun University, Gwangju 61452, Korea; lkj1998@chosun.ac.kr

**Keywords:** oxysterol, 25-hydroxycholesterol, apoptosis, oxidative stress, oxiapoptophagy

## Abstract

25-hydroxycholesterol (25-HC) is an oxysterol synthesized from cholesterol by cholesterol-25-hydroxylase during cholesterol metabolism. The aim of this study was to verify whether 25-HC induces oxiapoptophagy in fibroblasts. 25-HC not only decreased the survival of L929 cells, but also increased the number of cells with condensed chromatin and altered morphology. Fluorescence-activated cell sorting results showed that there was a dose-dependent increase in the apoptotic populations of L929 cells upon treatment with 25-HC. 25-HC-induced apoptotic cell death was mediated by the death receptor-dependent extrinsic and mitochondria-dependent intrinsic apoptosis pathway, through the cascade activation of caspases including caspase-8, -9, and -3 in L929 cells. There was an increase in the levels of reactive oxygen species and inflammatory mediators such as inducible nitric oxide synthase, cyclooxygenase-2, nitric oxide, and prostaglandin E2 in L929 cells treated with 25-HC. Moreover, 25-HC caused an increase in the expression of beclin-1 and microtubule-associated protein 1A/1B-light chain 3, an autophagy biomarker, in L929 cells. There was a significant decrease in the phosphorylation of protein kinase B (Akt) in L929 cells treated with 25-HC. Taken together, 25-HC induced oxiapoptophagy through the modulation of Akt and p53 cellular signaling pathways in L929 cells.

## 1. Introduction

Cholesterol is a lipid sterol that not only serves as an essential molecule for the survival and growth of mammalian cells, but also as a precursor of bile acids, sterol hormones, and vitamin D, which are also implicated in cell signaling processes [1]. Therefore, cholesterol homeostasis in both cellular and systemic concentrations is precisely regulated by the synthesis and transport systems in mammalian cells [2]. Generally, cholesterol is synthesized by the mevalonate pathway and then transported to destined membranes for structural and functional needs, upon uptake of low-density lipoprotein (LDL) from circulation through LDL-mediated endocytosis in every mammalian cell [3]. However, excess cholesterol is either exported from the cells by ATP-binding cassette transporters, or converted to cholesteryl esters with less cytotoxicity by acyl-coenzyme A:cholesterol acyltransferases and then stored in lipid droplets or secreted within lipoproteins [4]. Recent studies have revealed that dysregulation of cholesterol metabolism is not only closely associated with neurodegenerative disorders such as Alzheimer’s disease [5], Parkinson’s disease [6], Huntington’s disease [7], atherosclerosis [8], cardiovascular disease [9], and hypercholesterolemia [10], but also acts as a pathophysiological etiology of carcinogenesis in various types of cancers [11,12,13].

Oxysterols are oxidized derivatives of cholesterol that are generated by means of auto-oxidation by free radicals and reactive oxygen species (ROS), or formed enzymatically through a variety of enzymes that are involved in the synthesis of bile acid [14]. Although oxysterols generally exist at very low concentrations in vivo, they play a key role in various physiological functions, including development, regulation of immune cell responses, cell metabolism, and cell survival [14]. Recent studies reported that oxysterols induce a type of cell death called oxiapoptophagy, which is associated with oxidative stress, apoptosis, and autophagy [15].

25-hydroxycholesterol (25-HC) is an oxysterol synthesized from cholesterol by the catalytic reaction of cholesterol-25-hydroxylase (CH25H), which is an enzyme that adds a hydroxyl group at the 25-carbone of cholesterol (Figure 1) [16]. Recent studies have reported the multifunction of 25-HC in lipid metabolism, antiviral processes, inflammatory response, cell survival, and even apoptosis [16,17,18,19]. In the present study, we investigated 25-HC-induced oxiapoptophagy, a type of cell death associated with oxidative stress, apoptosis, and autophagy mediated by oxysterol, and its underlying cellular signaling pathways in L929 mouse fibroblast cells.

## 2. Results

### 2.1. 25-HC Induced Cell Death through an Increase in Cytotoxicity and Decrease in Cell Survival in L929 Cells

L929 cells were treated with 0, 1, 2.5, 5, 10, 20, and 40 μg/mL 25-HC for 48 h. Thereafter, a dimethyl thiazolyl diphenyl tetrazolium salt (MTT) assay was performed to investigate 25-HC-induced cytotoxicity in L929 cells. As shown in Figure 2a, 25-HC reduced the relative viability of L929 cells in a dose-dependent manner. The relative cell viabilities were 100.04 ± 5.2%, 80.4 ± 2.8%, 67.23 ± 2.4%, 24.7 ± 5.8%, 17.93 ± 1.1%, 12.67 ± 0.7%, and 9.09 ± 1.2% in L929 cells treated with 0, 1, 2.5, 5, 10, 20, and 40 μg/mL 25-HC, respectively. Upon cell LIVE/DEAD™ staining, nontreated control cells were stained green using cell-permeable Green Calcein AM, signifying live cells. In contrast, the number of dead cells stained with red fluorescence using ethidium homoidimer-1 increased in L929 cells treated with 1 and 5 μg/mL 25-HC for 48 h (Figure 2b). These data indicated that 25-HC induced cell death through a dose-dependent increase in cytotoxicity and decrease in the survival of L929 cells.

### 2.2. 25-HC-Induced Cell Death Involved in Apoptosis of L929 Cells

To determine whether 25-HC-induced L929 cell death is involved in apoptosis, representative apoptotic characteristics, such as chromatin condensation and morphological alteration, were investigated using 4′,6-diamidino-2-phenylindole dihydrochloride (DAPI) and hematoxylin and eosin (H&E) staining, respectively. As shown in Figure 3a, treatment with 1 and 5 μg/mL 25-HC not only decreased the total number of L929 cells, but also increased the number of cells with condensed chromatin. Furthermore, similar to the results of DAPI staining, H&E staining showed that there was not only a dose-dependent decrease in the number of L929 cells upon treatment with 25-HC, but also an increase in the number of cells with altered morphologies (Figure 3b). Taken together, these data suggested consistently that 25-HC-induced cell death is involved in apoptosis in L929 cells.

Next, to verify whether 25-HC-induced cell death is involved in apoptosis in L929 cells, fluorescence-activated cell sorting (FACS) analysis using propidium iodide (PI) and annexin-V was performed, as shown in Figure 4. The FACS results showed that 1 and 5 μg/mL 25-HC increased the apoptotic population of L929 cells in a dose-dependent manner. In particular, the populations of dead cells, including those in an early stage of apoptosis, late-stage apoptosis, and necrosis, were assessed to be 13.48%, 34.93%, and 77.9% in the L929 cells treated with 0, 1, and 5 μg/mL 25-HC for 48 h, respectively, in a dose-dependent manner. These data indicated that 25-HC induced apoptosis in L929 cells.

### 2.3. 25-HC-Induced Cell Death Is Mediated by FasL-Triggered Death Receptor-Dependent Extrinsic and Mitochondria-Dependent Intrinsic Apoptosis Pathways through the Cascade Activation of Caspases in L929 Cells

To determine the cellular signaling pathway associated with 25-HC-induced cell death, cells were treated with 1 and 5 μg/mL 25-HC for 48 h. Thereafter, total proteins were extracted, and Western blot was performed, as shown in Figure 5. 25-HC not only upregulated the expression of Fas ligand (FasL), a representative death ligand that induces the death receptor-mediated extrinsic apoptosis pathway, but also increased the levels of cleaved caspase-8, a downstream target molecule of FasL, in L929 cells (Figure 5a). In addition, the expression of BH3 interacting-domain death agonist (Bid), a proapoptotic factor that initiates the mitochondria-dependent intrinsic apoptosis pathway after the cleavage of Bid to truncated Bid (tBid), was reduced by cleaved caspase-8, as shown in Figure 5b. Furthermore, there was a decrease in the expression of antiapoptotic factors such as B-cell lymphoma 2 (Bcl-2) and B-cell lymphoma-extra large (Bcl-xL) in L929 cells treated with 25-HC. In contrast, 25-HC increased the expression of proapoptotic factors such as Bcl-2-associated X protein (Bax) and Bcl-2-associated agonist of cell death (Bad) in L929 cells. Moreover, 25-HC decreased the expression of cleaved caspase-9 in L929 cells. Sequentially, both cleaved caspase-8 and -9 initiated the cleavage of pro-caspase-3 in L929 cells treated with 25-HC. Hence, there was an increase in the expression and activation of cleaved caspase-3 in L929 cells treated with 25-HC, as shown in Figure 5c,d. Finally, cleaved caspase-3 induced the cleavage of poly (ADP-ribose) polymerase (PARP) to cleaved PARP in L929 cells treated with 25-HC. Taken together, these data indicated that 25-HC-induced cell death is mediated by death receptor-mediated extrinsic and mitochondria-dependent intrinsic apoptosis pathways, depending on the cascade activation of caspases in L929 cells.

### 2.4. 25-HC Upregulated the Expression and Production of ROS and Inflammatory Mediators in L929 Cells

5-(and-6)-carboxy-2′,7′-dichlorodihydrofluorescein diacetate (DCFDA) staining was performed to investigate whether 25-HC induced ROS production in L929 cells. As shown in Figure 6a, there was a significant increase in the number of cells stained with DCFDA in L929 cells treated with 1 and 5 μg/mL 25-HC for 48 h. The FACS results showed that the relative intensity of ROS was assessed by 199.41 ± 13.3% and 301.51 ± 15.85% in the L929 cells treated with 1 and 5 μg/mL 25-HC, respectively, compared with that in untreated control (100.18 ± 7.04%) (Figure 6b).

Furthermore, there was a significant increase in the expression and production of inflammatory mediators such as inducible nitric oxide synthase (iNOS), cyclooxygasense-2 (COX-2), and prostaglandin E_2_ (PGE_2_) in L929 cells treated with 1 and 5 μg/mL 25-HC for 48 h, as shown in Figure 6c,d. Taken together, these data indicated that 25-HC induced ROS and inflammatory mediators in L929 cells.

### 2.5. 25-HC-Induced Apoptosis Is Accompanied by p53-Dependent Autophagy through the Suppression of Akt Phosphorylation in L929 Cells

As shown in Figure 7, the alteration of biomarkers associated with autophagy was investigated using Western blot and laser confocal scanning microscopy to determine whether 25-HC induced oxiapoptophagy in L929 cells. The results of Western blot showed that 25-HC increased autophagy biomarkers such as beclein-1 and microtubule-associated protein light chain 3 (LC-3) in L929 cells (Figure 7a). Furthermore, confocal scanning microscopic images showed expression of beclin-1 in the cytosol of L929 cells treated with 25-HC (Figure 7b).

Furthermore, 25-HC significantly decreased the phosphorylation of Akt, a representative suppressor of autophagy [20] in L929 cells. However, the expression of p53, a molecule associated with the inhibition of autophagy [21], was significantly increased in L929 cells treated with 25-HC, as shown in Figure 7c. Taken together, these data indicated that 25-HC-induced apoptosis is accompanied by p53-dependent autophagy through the suppression of Akt phosphorylation in L929 cells.

## 3. Discussion

A variety of lipid components and their derivatives play critical roles in physiological processes and act as structural components, energy stores, and signaling molecules in physiological and pathophysiological conditions [22]. Cellular membranes, which are mainly composed of lipids, including cholesterol, act as a guide for cellular pathways and functionalities such as cell polarization and trafficking between the surface and intracellular membrane, signal transduction, cell growth, migration, and even cell death [23]. In particular, cholesterol, the major sterol present in animal tissues, not only serves as an essential component for plasticity of plasma membranes, but also has various other physiological functions, including as a precursor of steroid hormones such as estrogens and androgens, bile acid, and vitamin D [24]. However, oxidative stress results in an imbalance between free radicals and antioxidants and is closely associated with damage to fatty tissue, DNA, and proteins, which can lead to several diseases, including diabetes, atherosclerosis, low-grade chronic inflammation, hypertension, heart disease, neurodegenerative diseases, and even cancer [25,26,27]. In addition, oxidative stress and low-grade chronic inflammation caused by aging and metabolic syndrome contribute to the synthesis of oxysterols, an oxidized form of cholesterol, generated from cholesterol by auto-oxidation, enzymatic processes, or both [28,29,30].

Recently, Russo et al. reported that *ch25h* mRNA and 25-HC are upregulated in a high-fat diet mouse model. On the other hand, there is a reduction in inflammatory responses in *ch25h*-knockout mice [31]. These results indicate that CH25H and 25-HC act as physiological mediators between inflammation and obesity. In particular, CH25H, known as cholesterol-25-monooxygenase, is mainly localized in the endoplasmic reticulum and Golgi apparatus and catalyzes the synthesis of 25-HC from cholesterol [32]. Although the expression of CH25H is low and stable in most organs under normal physiological conditions, it is significantly increased by viral infections, Toll-like receptor agonists, and under inflammatory conditions [33]. Furthermore, Ichikawa et al. have reported that 25-HC is closely associated with the pathogenesis of chronic inflammatory lung diseases through fibroblast-mediated tissue remodeling of human fetal fibroblasts HEL-1 cells and U2-OS cells [34]. Thus, the level of 25-HC is generally upregulated under pathological and inflammatory conditions. Recent studies have reported that 25-HC is involved in a variety of physiological processes, including cholesterol homeostasis, inflammatory response, immune cell migration, carcinogenesis in breast and ovarian cancer cells, and a broad antiviral function in both enveloped and non-enveloped viruses [32].

Recent studies have reported that at cytotoxic concentrations, some oxysterols such as 5,6-epoxycholesterol, 7-ketocholesterol (7KC), 7β-HC, and 24(S)-HC, induce oxiapoptophagy in different cell types from different species [15]. Although Sankhavaram et al. reported that 25-HC-induced apoptosis was closely associated with the calcium ion flux into cultured fibroblast cells [35], 25-HC-induced oxiapoptophagy has not been reported in fibroblasts. Recent studies have shown that 25-HC is closely associated with apoptosis in a dose-dependent manner [18,19,36,37]. Ayala-Torres et al. reported apoptotic characteristics, including breakdown of DNA, morphological alteration, and attenuation of apoptosis by a pan-caspase inhibitor Z-VAD-FMK in the human leukemic cell line CEM-C7, upon treatment with 300 nM 25-HC; the effect varied in a time-dependent manner [36]. Ares et al. reported a time-dependent increase in apoptotic characteristics such as chromatin condensation and activation of caspase-3 in human aortic smooth muscle cells treated with 5 μg/mL 25-HC [38]. Furthermore, Ares et al. reported that 25-HC-induced apoptosis was synergistically increased upon co-treatment with proinflammatory cytokines such as 10 ng/mL tumor necrosis factor α and 20 ng/mL interferon-γ in human aortic smooth muscle cells, but significantly inhibited by Ca^2+^ entry blockers, such as verapamil or nifedipine [38]. Travert et al. reported that 25-HC not only reduced cholesterol biosynthesis in adult rat Leydig cells, but also increased cytotoxicity through the induction of apoptotic cell death accompanied by an increase in DNA fragmentation, condensed chromatin, and expression of caspase-3 [39]. Moreover, Choi et al. reported that mitochondria-dependent intrinsic apoptosis, which is mediated by the cascade activation of the caspase-9-caspase-3 axis through the activation of JNK and glycogen synthase kinase-3β cellular signaling pathway in pheochromocytoma PC12 cells treated with 2 and 10 nM 25-HC for 28 h [17]. Recent studies have reported that 25-HC induces apoptosis in various types of cells, such as oligodendrocyte cell line 158N [37], primary rat chondrocytes [18], and human head and neck squamous cell carcinoma FaDu cells [19]. More recently, Olivier et al. reported that 25-HC induces both P2X7-dependent pyroptosis and caspase-dependent apoptosis in a human skin model using keratinocytes [40]. Olivier et al. reported that the level of 25-HC not only increased in keratinocytes after UV irradiation but was also associated with skin degeneration [40]. Taken together, these results suggest that 25-HC is closely associated with apoptotic cell death [40]. However, it remains controversial in other types of cells, and the underlying mechanisms are poorly understood.

Recent studies have reported that some of oxysterols, including 5,6-Epoxycholesterol isomers and 25-HC, induce cell death through the death receptor-mediated extrinsic and mitochondria-dependent intrinsic apoptosis pathways [19,41]. Especially mitochondria-dependent intrinsic apoptosis is closely associated with the loss of mitochondrial membrane potential (ΔΨm) that is resulted in mitochondrial membrane permeabilization and the production of ROS [41,42]. Sequentially, highly increasing ROS also act as inducers of apoptotic cell death in various types of cells, including fibroblast cells [41,42,43]. In addition, the increasing ROS induce the increase of autophagosome [44]. Sequentially, LC3 is translocated to autophagosomes for converting LC3-I to LC3-II, which is a robust biomarker of autophagy, to induce autophagy [45]. Hence, some oxysterols-induced cell death accompanied by oxidative stress, apoptosis, and autophagy is called oxiapoptophagy [15].

In the present study, we demonstrated that the viability of the mouse fibroblast cell line L929 decreased in a dose-dependent manner upon treatment with 1–40 μg/mL 25-HC, as shown in Figure 2a. Sequentially, not only did the number of cells decrease in L929 cells treated with 1–5 μg/mL 25-HC, but the number of dead cells stained with red fluorescence by ethidium homodimer-1 also increased in a dose-dependent manner (Figure 2b). Furthermore, our results using DAPI (Figure 3a) and H&E (Figure 3b) staining showed that apoptotic characteristics such as chromatin condensation and alterations in morphology were revealed in L929 cells treated with 1–5 μg/mL 25-HC. FACS results using PI and Annexin-V staining showed that there was an increase in the apoptotic populations in L929 cells, as shown in Figure 4. Moreover, 25-HC-induced apoptosis was mediated by the FasL-induced death receptor-dependent extrinsic apoptosis mediated by the cascade activation of the caspase-8-caspase-3-PARP axis and mitochondria-dependent intrinsic apoptosis mediated by the cascade activation of the caspase-8-Bid-caspase-9-caspase-3-PARP axis in L929 cells (Figure 5).

Oxidative stress is not only associated with overproduction of ROS, lipid peroxidation, and protein carbonylation, but also closely associated with the synthesis of oxysterol through enzymatic, non-enzymatic, and auto-oxidation of cholesterol [15]. Notably, oxysterols such as 7KC, 7β-HC, and 24(S)-HC induce overproduction of ROS in various types of cells, such as murine oligodendrocyte 158N, murine microglial cells BV-2, and murine neuroblastoma cell line N2a cells [15,45,46]. Hence, oxysterol-mediated oxidative stress-apoptosis is a characteristic of oxiapoptophagy. In the present study, 25-HC not only increased ROS production, but also upregulated COX-2 expression in L929 cells (Figure 6). These data indicated that 25-HC-induced apoptosis was accompanied by oxidative stress in L929 cells.

Autophagy, also known as autophagocytosis, is a term for the degradation of cytoplasmic components within lysosomes, which is mediated by autophagosomes to engulf a portion of the cytoplasm [47,48]. The molecular mechanism of autophagy is not only mediated by several conserved autophagy-related proteins, such as belcin-1 and LC3, but also by the Akt and p53 cellular signaling pathways [47]. Generally, belcin-1 and LC3 are required to initiate the formation of autophagosomes in mammals. Furthermore, in the regulation of mTOR cellular signaling mediated autophagy, the Akt cellular signaling pathway is involved in the inhibition of autophagy [49]. Moreover, p53 is a critical mediator of damage-induced apoptosis and directly induces autophagy damage-regulated autophagy modulator (DRAM), which is a lysosomal integral membrane protein that contributes to the accumulation of autophagosomes in a dependent manner to execute full cell death [50]. As shown in Figure 7, the expression of autophagy-related proteins, such as beclin-1 and LC3, was upregulated in the L929 cells treated with 25-HC. In particular, the conversion of LC3-I to LC3-II through the conjugation of phosphatidylethanolamine indicates the formation of autophagosome [51]. In present study, both LC3-I and LC3-II were increased in the L929 cells treated with 25-HC. Furthermore, the expression of LC3-II was higher than that of LC3-I in the L929 cells treated with 25-HC. Hence, these data indicate that 25-HC induces autophagy through the formation of autophagosome in L929 cells.

Furthermore, the phosphorylation of Akt was reduced by 25-HC in L929 cells. In contrast, there was a significant increase in the expression of p53 in L929 cells treated with 25-HC. Taken together, these data indicated that 25-HC-induced apoptosis was involved in autophagy through the suppression of the mTOR-Akt cellular signaling pathway and p53-mediated DRAM axis in L929 cells. Therefore, these data indicated that 25-HC induced oxiapoptophagy accompanied by oxidative stress, apoptosis, and autophagy in L929 cells.

Recently, Kim et al. reported that the level of 25-HC in the serum is approximately 4.27 ± 1.18 ng/mL in normal humans, whereas it is significantly higher, approximately 5.39 ± 1.94 ng/mL, in patients with amyotrophic lateral sclerosis [52]. Furthermore, Russo et al. reported a level of 25-HC in the range of 27.4–437 ng/g in human visceral adipose tissue harvested from bariatric surgery patients [31]. A limitation of the present study is that we used a high concentration of 25-HC to induce oxiapoptophagy in L929 cells, as compared to what has been reported in normal humans and patients who underwent bariatric surgery. Hence, our future studies will focus on verifying 25-HC-induced physiological effects including oxiapoptophagy and its pathophysiological mechanisms in animal models with low-grade chronic inflammation mediated by aging and metabolic syndromes.

In conclusion, in the present study, we demonstrated 25-HC-induced oxiapoptophagy in fibroblasts, the most common connective tissue cells that act as a structural stroma through the synthesis of extracellular matrix and collagen. We believe that 25-HC is a key factor associated with low-grade chronic inflammation caused by aging and metabolic syndrome, which is involved in the degeneration of tissues composed of fibroblasts.

## 4. Materials and Methods

### 4.1. Cell Culture

L929 cells, fibroblast cells derived from mouse subcutaneous connective tissue, were obtained from the Korean Cell Line Bank (Seoul, Korea) and maintained in minimum essential medium containing 10% fetal bovine serum (Life Technologies, Grand Island, NY, USA) and antibiotics (50 U/mL penicillin and 50 μg/mL streptomycin) in a humidified incubator at 37 °C with 5% CO_2_.

### 4.2. Cell Viability Assay

Cells (1 × 10^5^ cells/mL) were cultured in 96-well culture plates and treated with 0, 1, 2.5, 5, 10, 20, and 40 μg/mL 25-HC for 48 h. After adding MTT solution, the cells were further cultured for 4 h. After incubation, the formed MTT crystals were suspended completely in dimethyl sulfoxide (Sigma-Aldrich, St. Louis, MO, USA), and the resulting absorbance at 570 nm was measured using an Epoch microplate spectrophotometer (BioTek, Winooski, VT, USA).

### 4.3. Cell Survival Assay

Cells (1 × 10^5^ cells/mL) were cultured in 8-well chamber slides (Nunc^®^ Lab-Tek^®^ Chamber Slide™ system, Sigma-Aldrich) and treated with 0, 1, and 5 μg/mL 25-HC for 48 h. Thereafter, LIVE/DEAD™ cell assays were performed using a LIVE/DEAD™ cell assay kit (Thermo Fisher Scientific, Rockford, IL, USA), which comprises Green Calcein AM (to stain the live cells with green fluorescence) and ethidium homodimer-1 (to stain the dead cells with red fluorescence), according to the manufacturer’s instructions. The cells were imaged using a fluorescence microscope (Eclipse TE2000; Nikon Instruments Inc., Melville, NY, USA).

### 4.4. Nuclear Staining

Cells (1 × 10^5^ cells/mL) were cultured in an 8-well chamber slide and treated with 0, 1, and 5 μg/mL 25-HC for 48 h. After cultivation, the cells were rinsed three times with phosphate-buffered saline (PBS; Sigma-Aldrich) and then stained with DAPI (Sigma-Aldrich). Nuclear condensation was observed and imaged using a fluorescence microscope (Eclipse TE200).

### 4.5. H&E Staining

H&E staining was performed to investigate the morphological changes induced by 25-HC in cells. Briefly, the cells (1 × 10^5^ cells/mL) were cultured in an 8-well chamber slide and allowed to adhere to the well overnight. The cultured cells were then treated with 0, 1, and 5 μg/mL 25-HC for 48 h at 37 °C. Thereafter, the cells were rinsed three times with PBS at 4 °C and fixed with 4% paraformaldehyde for 30 min at 4 °C. H&E staining was performed to observe the morphological changes in the cells. The cells were observed and imaged using a microscope (Leica DM750; Leica Microsystems, Heerbrugg, Switzerland).

### 4.6. FACS Analysis

Cells (1 × 10^5^ cells/mL) were cultured in 6-well culture plates and then treated with 0, 1, and 5 μg/mL 25-HC for 48 h. Thereafter, the cells were collected, washed with ice-cold potassium PBS, and resuspended in binding buffer (BD Biosciences, San Diego, CA, USA). Annexin V-fluorescein isothiocyanate (Annexin V-FITC) and PI (Cell Signaling Technology, Danvers, MA, USA) were then added to the cells and incubated for 15 min at 37 °C. Changes in the apoptotic population were analyzed using BD Cell Quest^™^ version 3.3 (Becton Dickinson, San Jose, CA, USA).

### 4.7. Western Blot

Cells (1 × 10^5^ cells/mL) were cultured in 6-well culture plates and then treated with 0, 1, and 5 μg/mL 25-HC for 48 h. Thereafter, total proteins were extracted from the L929 cells using cell lysis buffer (Cell Signaling Technology), according to the manufacturer’s instructions. Protein concentration was determined using a bicinchoninic acid protein assay (Thermo Fisher Scientific). Equal amounts of each protein sample were electrophoresed on a 10% sodium dodecyl sulfate polyacrylamide gel and subsequently transferred to a polyvinylidene fluoride (PVDF) membrane (Millipore, Burlington, MA, USA) at 4 °C. Thereafter, the PVDF membrane was blocked using 5% (*v*/*v*) bovine serum albumin (BSA; Sigma-Aldrich) prepared in Tris-buffered saline with Tween 20 (TBS-T; Santa Cruz Biotechnology Inc., Dallas, TX, USA) and then incubated with the primary antibodies diluted 1:1000 in TBS-T containing 5% (*v*/*v*) BSA at 4 °C for 12 h. The immunoreactive bands were visualized using the ECL System (Amersham Biosciences, Piscataway, NJ, USA) and exposed on radiographic film or MicorChemi 4.2 (Dong-Il Shimadzu Corp., Seoul, Korea).

### 4.8. NO Measurement

Cells (1 × 10^5^ cells/mL) were cultured in 6-well culture plates and treated with 0, 1, and 5 μg/mL 25-HC for 48 h. Thereafter, the conditioned medium (100 μL) was reacted with 100 μL sulfanilamide and *N*-1-naphthylethylenediamine dihydrochloride. Absorbance at 540 nm was measured using a spectrophotometer (Epoch spectrophotometer).

### 4.9. PGE_2_ Production Measurement

Cells (1 × 10^5^ cells/mL) were cultured in 6-well culture plates and then treated with 0, 1, and 5 μg/mL 25-HC for 48 h. Thereafter, PGE_2_ production was measured using a Parameter™ PGE_2_ assay kit (Thermo Fisher Scientific) according to the manufacturer’s protocol.

### 4.10. Detection of ROS

Cells (1 × 10^5^ cells/mL) were cultured in an 8-well chamber slide and allowed to adhere to the well overnight. Cultured cells were treated with 0, 1, or 5 µg/mL 25-HC for 48 h at 37 °C. Thereafter, DCFDA staining was performed to detect ROS, according to the manufacturer’s instructions. Cells were imaged using fluorescence microscopy (Eclipse TE200). In addition, the intensity of ROS was measured by FACS using DCFDA according to the manufacturer’s instructions.

### 4.11. Caspase-3/-7 Activity Assay

Cells (1 × 10^5^ cells/mL) were cultured in an 8-well chamber slide and allowed to adhere to the well overnight. Cultured cells were treated with 0, 1, or 5 µg/mL 25-HC for 48 h at 37 °C. Thereafter, the activity of caspase-3/7 was assessed using the cell-permeable fluorogenic substrate PhiPhiLux-G_1_D_2_ (OncoImmunin Inc., Gaithersburg, MD, USA) according to the manufacturer’s instructions and imaged using fluorescence microscopy (Eclipse TE200).

### 4.12. Laser Confocal Scanning Microscopic Analysis

Cells (1 × 10^5^ cells/mL) were cultured in an 8-well chamber slide and allowed to adhere to the well overnight. Cultured cells were treated with 0, 1, or 5 µg/mL 25-HC for 48 h at 37 °C. After the treatments, the cells were fixed with 1% paraformaldehyde, permeabilized in 0.2% Triton™ X-100, and extensively washed with PBS. Nonspecific signals were blocked using normal goat serum. After multiple washes, the cells were incubated with rabbit anti-belcin-1 antibodies, followed by incubation with FITC-conjugated goat anti-rabbit IgG (Thermo Fisher Scientific, Waltham, MA, USA) overnight at 4 °C. The stained cells were imaged using a laser confocal scanning microscope system (Leica Microsystems, Wetzlar, Germany) at the Gwangju branch of the Korea Basic Science Institute (Gwangju, Korea).

### 4.13. Statistical Analysis

The experimental data are presented as mean  ±  standard deviation from at least three independent experiments and were compared using analysis of variance, followed by Student’s *t*-test. Statistical significance was set at *p* < 0.05.

## Figures and Tables

**Figure 1 molecules-27-00199-f001:**
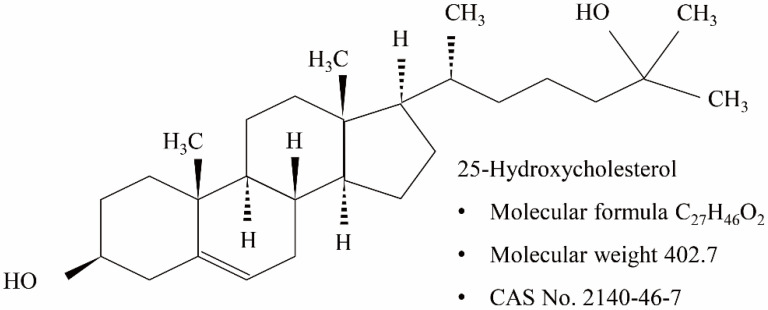
Chemical structure of 25-hydroxycholesterol.

**Figure 2 molecules-27-00199-f002:**
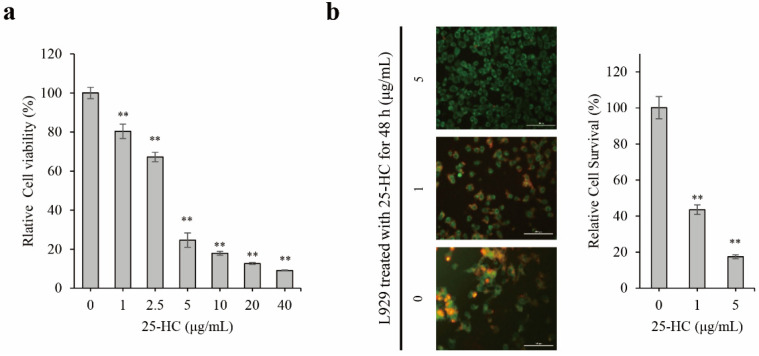
25-HC induced cell death by increasing cytotoxicity and decreasing cell survival in L929 cells. (**a**) 25-HC decreased the viability of L929 cells in a dose-dependent manner. An MTT assay was performed to measure the viability of L929 cells treated with 0–40 μg/mL 25-HC for 48 h. (**b**) 25-HC decreased the survival of L929 cells. A LIVE/DEAD™ assay was performed using Green Calcein AM to stain the live cells with green fluorescence and ethidium homodimer-1 to stain the dead cells with red fluorescence in order to assess the survival of L929 cells treated with 1 and 5 μg/mL 25-HC for 48 h. Relative cell survival was calculated by [number of live cells/(number of live cells + number of dead cells) × 100] and was presented as histogram. Results are mean ± SD of three independent experiments (*n* = 3), ** *p* < 0.01.

**Figure 3 molecules-27-00199-f003:**
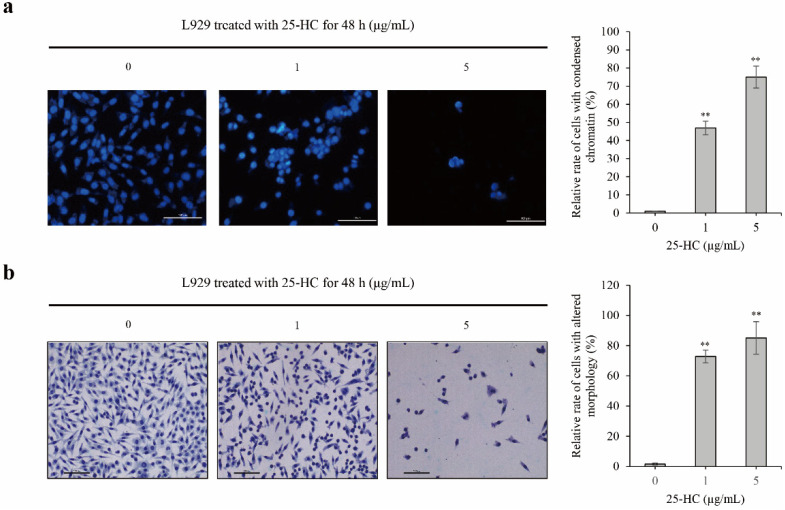
25-HC not only decreased the number of cells, but also increased the number of L929 cells with apoptotic characteristics, such as condensed chromatin and altered morphologies. L929 cells were cultured on 8-well chamber slides and treated with 1 and 5 μg/mL 25-HC for 48 h. DAPI (**a**) and H&E staining (**b**) were performed to observe chromatin condensation and morphological alteration, respectively. Thereafter, relative rates of cells with condensed chromatin or altered morphology were calculated by [number of cells with condensed chromatin or altered morphology/total number of cells × 100] and are presented as a histogram. Results are mean ± SD of three independent experiments (*n* = 3), ** *p* < 0.01. (**a**) 25-HC increased the number of L929 cells with condensed chromatin in a dose-dependent manner. (**b**) 25-HC increased the number of L929 cells with altered morphology.

**Figure 4 molecules-27-00199-f004:**
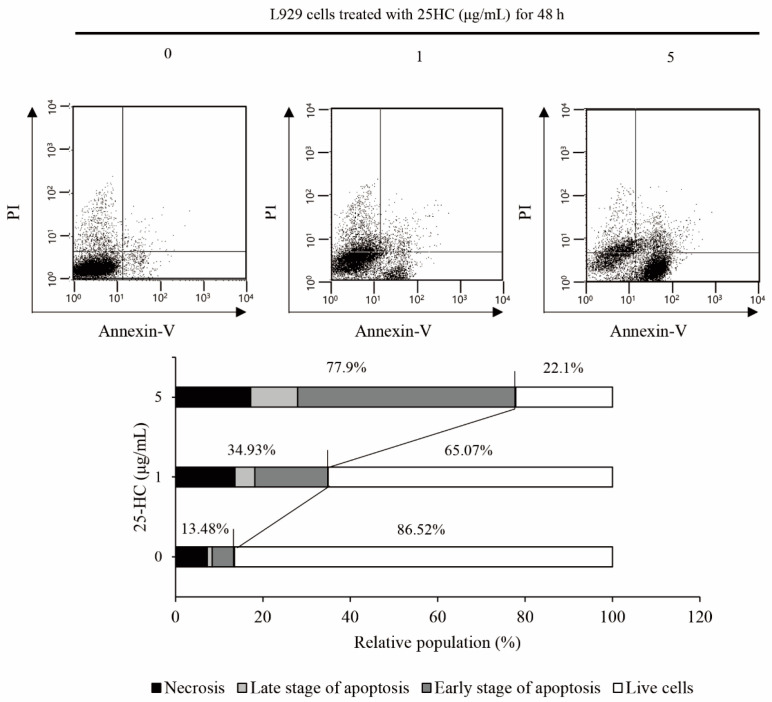
25-HC increased the apoptotic population of L929 cells in a dose-dependent manner. L929 cells were treated with 1 and 5 μg/mL 25-HC for 48 h. Thereafter, FACS using PI and Annexin V was performed to analyze the apoptotic population.

**Figure 5 molecules-27-00199-f005:**
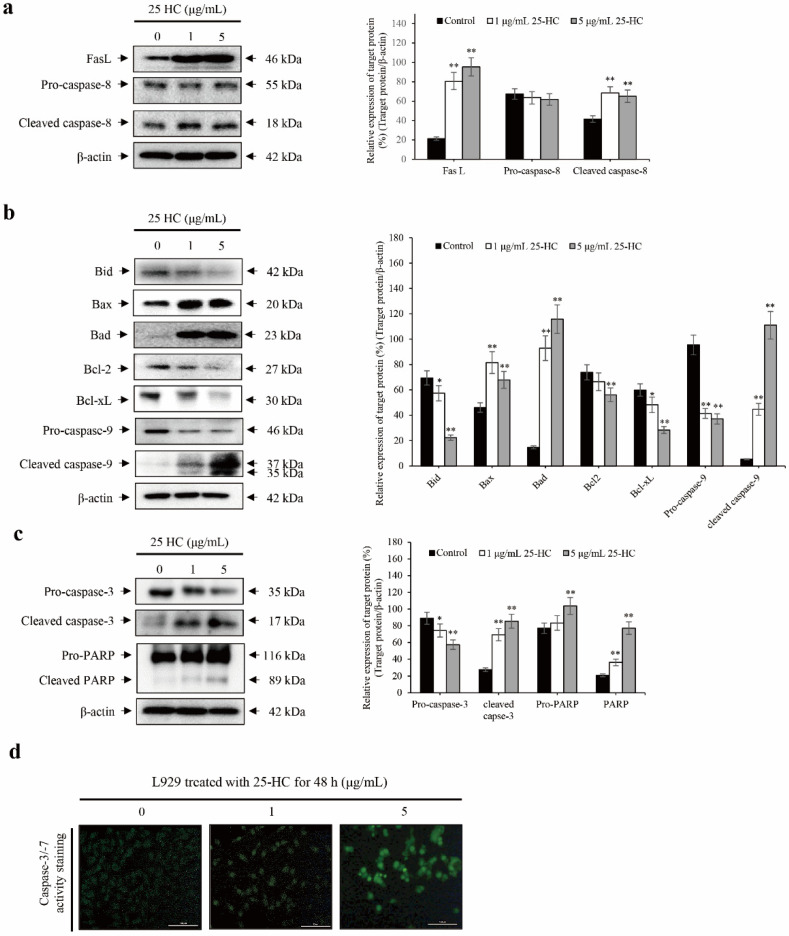
25-HC-induced cell death was mediated by the cascade activation of caspases through the death receptor-mediated extrinsic and mitochondria-dependent intrinsic apoptosis pathways in L929 cells. L929 cells were treated with 1 and 5 μg/mL 25-HC for 48 h. Thereafter, total proteins were extracted and electrophoresed on the SDS-PAGE gels to perform Western blot using specific antibodies associated with pro- and antiapoptotic factors. The results were expressed as the relative ratio of target protein/β-actin, * *p* < 0.05 and ** *p* < 0.01. β-actin was used as the internal control. In addition, the activity of caspase-3 was assessed by staining the cell-permeable fluorogenic substrate PhiPhiLux-G_1_D_2_ in L929 cells treated with 1 and 5 μg/mL 25-HC for 48 h. L929 cells were treated with 1 and 5 μg/mL 25-HC for 48 h. (**a**) Thereafter, 25-HC upregulated the expression of FasL and caspase-8, which are proapoptotic factors associated with death receptor-mediated extrinsic apoptosis in L929 cells. (**b**) 25-HC not only decreased the expression of antiapoptotic factors such as Bcl-2 and Bcl-xL, but also increased the expression of proapoptotic factors such as Bax, Bad, and caspase-9 in the mitochondria-dependent intrinsic apoptosis pathway in L929 cells. (**c**) Activated caspase-8 and -9 induced cell death through activation of caspase-3 and PARP in L929 cells. (**d**) There was a dose-dependent increase in the activity of caspase-3 in L929 cells treated with 25-HC.

**Figure 6 molecules-27-00199-f006:**
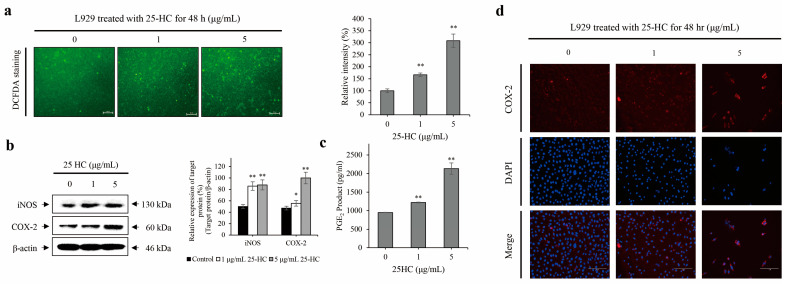
25-HC upregulated the expression and production of ROS and inflammatory mediators in L929 cells. L929 cells were treated with 1 and 5 μg/mL 25-HC for 48 h. Thereafter, DCFDA staining and intensity (**a**), Western blot (**b**) for iNOS and COX-2, PGE_2_ (**c**) assay, and immunocytochemistry using COX-2 antibody (**d**) were performed to investigate the upregulation of ROS and inflammatory mediators in L929 cells treated with 25-HC. (**a**) 25-HC increased the production of ROS in L929 cells. (**b**–**d**) The expression and production of inflammatory mediators including iNOS, COX-2, and PGE_2_ were upregulated in L929 cells treated with 25-HC. The results of Western blot were expressed as the relative ratio of target protein/β-actin. β-actin was used as the internal control. Significance was represented as * *p* < 0.05, ** *p* < 0.01.

**Figure 7 molecules-27-00199-f007:**
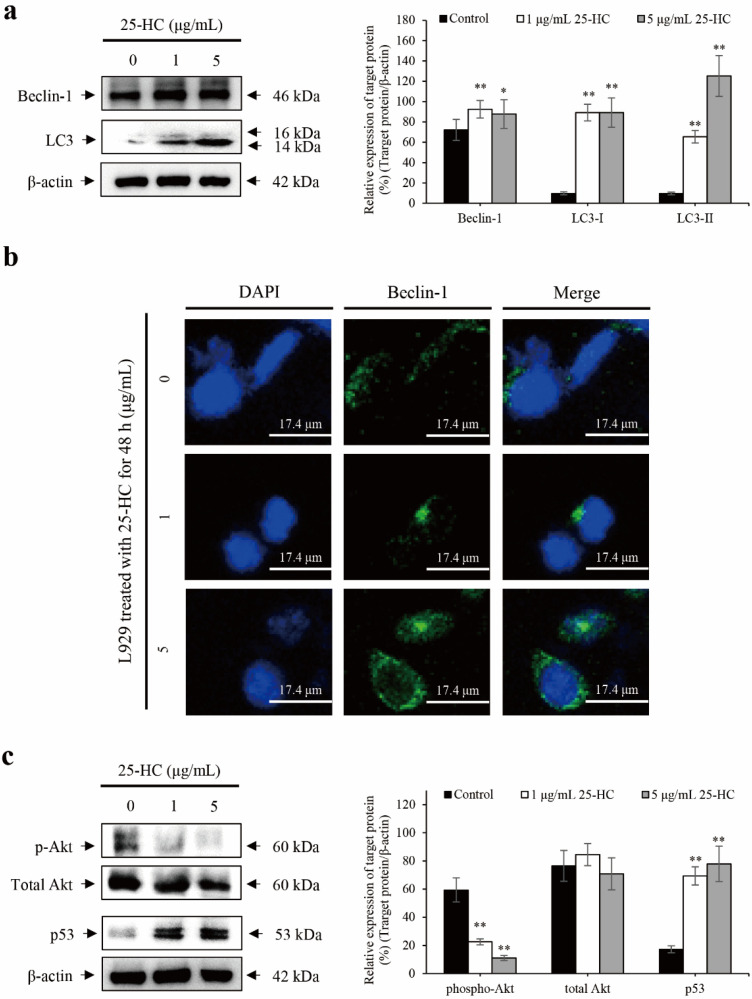
25-HC-induced apoptosis was accompanied by p53-dependent autophagy through the suppression of Akt phosphorylation in L929 cells. L929 cells were treated with 1 and 5 μg/mL 25-HC for 48 h. Thereafter, Western blot (**a**) using autophagy biomarkers such as beclin-1 and LC3, immunocytochemistry (**b**) using beclin-1 antibody and DAPI for staining nuclei and Western blot (**c**) using antibodies of cellular signaling molecules associated with autophagy such as Akt and p53 were performed to investigate 25-HC-induced autophagy and the underlying cellular signaling pathways in L929 cells. (**a**) Treatment with 25-HC caused an increase in the expression of autophagy biomarkers such as belcin-1 and LC3 in L929 cells. (**b**) There was an increase in the expression of beclin-1 in L929 cells treated with 25-HC. (**c**) 25-HC not only decreased the phosphorylation of Akt, but also increased the expression of p53 in L929 cells. The results of Western blot were expressed as the relative ratio of target protein/β-actin, * *p* < 0.05 and ** *p* < 0.01. β-actin was used as the internal control.

## Data Availability

The data presented in this study are available on request from the corresponding author.

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
