# Peer review of "25-Hydroxycholesterol-Induced Oxiapoptophagy in L929 Mouse Fibroblast Cell Line"

_molecules, 2021, doi:10.3390/molecules27010199_

Round 1
Reviewer 1 Report
25-hydroxycholesterol (25-HC) is an oxysterol synthesized from cholesterol by cholesterol-25-hydroxylase during cholesterol metabolism. This manuscript aims to verify whether 25-HC induces oxiapoptophagy in fibroblasts. The results in the manuscript demonstrated that 25-HC induced oxiapoptophagy through the modulation of Akt and p53 cellular signaling pathways in L929 cells. The finding is attractive to the readers in the related research fields. However, the data presenting could be improved to be more friendly to readers.
Some concerns:
- The gating of Annexin V/PI analysis in Figure 4 cannot not be used to distinguish early and late apoptotic cells.
- There are no quantitative results of Western blots. It is highly recommended to include triple repeats results with statistical results.
- Statistical results should be provided for Fig 2b, Fig 3, Fig 5, Fig 6abd and Fig7.
- Oxiapoptophagy is focused on how cell responses to stress. However, the link in the manuscript can be enhanced.
- Flow cytometry analysis of DCFDA and DHE staining would provide more information. Additionally, flow cytometry will also aid to verify the microscopy image result of Fig 6a.
- LC3 I and LC3 II ratio should examined to demonstrate the autophagy status.
- The authors should discuss references related to 25-hydroxycholesterol and fibroblast, such as:
- 25-hydroxycholesterol promotes fibroblast-mediated tissue remodeling through NF-κB dependent pathway (PMID: 23485764)
- Metabolism of 25-hydroxycholesterol in mammalian cell cultures. Side-chain scission to pregnenolone in mouse L929 fibroblasts (PMID: 2794780)
Arachidonate metabolism and the signaling pathway of induction of apoptosis by oxidized LDL/oxysterol (PMID: 11590225).
Author Response
Thank you for your encouragement to resubmit a revised version of our manuscript entitled, “25-Hydroxycholesterol-induced oxiapoptophagy in L929 mouse fibroblast cell line”. We thank you and the reviewers for giving us the opportunity to revise our manuscript and for providing valuable comments and suggestions. In response to the reviewers’ comments, we have revised our original manuscript to improve the quality of our findings and strengthen our original conclusions.
Please see the attachment.

Reviewer 2 Report
In the current manuscript, authors studied the role of 25-HC induction oxiapoptophagy in fibroblasts. Mounting evidence suggests that 25-HC decreased the survival of L929 cells, and also increased the number of cells with condensed chromatin and altered morphology. Authors found that fluorescence-activated cell sorting there was a dose-dependent increase in the apoptotic populations of L929 cells upon treatment with 25-HC. 25-HC-induced apoptotic cell death was mediated by death receptor-dependent extrinsic and mitochondria-dependent intrinsic apoptosis pathway, through the cascade activation of caspases including caspase-8, -9, and -3 in L929 cells. These observations are important and worth reporting.
Concerns: Quantifications of western blots and immunofluorescence are missing in Figures 5-7. It is important to present the quantification data with P values and statistical significance -
Author Response

(The authors gave the same response as above.)

Round 2
Reviewer 1 Report
The authors responded to the previous comments points by points. However, there are issues remained to be clarified.
- The FACS results in Figure 4 did not demonstrate a significant induced apoptotic cell. It appeared more like a dosage-dependent event. It is reviewer's opinion that the authors should provide control (such as solvent only) result to explain the gating strategy. Additionally, it's better to provide molarity instead of µg/mL for 25-HC in the manuscript.
- When providing the quantitative results, such as (number of cells with condensed chromatin or altered morphology/total number of cells) ×
100, it will be more convincible to provide the "n" value. - ROS included different species. DCFDA is used to detect the production of H2O2 in live cells (esterases), DHE is used to detect superoxide. In revised Figure 6, the readers would not be able to under the quantitative fluorescence intensity is calculated based on microscopy or plate reader.
- The western blot of LC3 in Figure 7 could be either mislabeled or wrong image.
Author Response
Thank you for your encouragement to resubmit a revised version of our manuscript entitled, “25-Hydroxycholesterol-induced oxiapoptophagy in L929 mouse fibroblast cell line”. We thank you and the reviewers for giving us the opportunity to revise our manuscript and for providing valuable comments and suggestions. In response to the reviewers’ comments, we have revised our original manuscript to improve the quality of our findings and strengthen our original conclusions. Please see attached file
